

# Bose-Einstein condensate in an elliptical waveguide

**Luca Salasnich**

Dipartimento di Fisica e Astronomia "G. Galilei" and QTech, Università di Padova,
Via F. Marzolo 8, I-35131, Padova, Italy
Istituto Nazionale di Fisica Nucleare, Sezione di Padova,
Via Marzolo 8, 35131 Padova, Italy
Istituto Nazionale di Ottica del Consiglio Nazionale delle Ricerche,
Unita di Sesto Fiorentino, Via Nello Carrara 2, 50019 Sesto Fiorentino, Italy

## Abstract

We investigate the effects of spatial curvature for an atomic Bose-Einstein condensate confined in an elliptical waveguide. The system is well described by an effective 1D Gross-Pitaevskii equation with a quantum-curvature potential, which has the shape of a double-well but crucially depends on the eccentricity of the ellipse. The ground state of the system displays a quantum phase transition from a two-peak configuration to a one-peak configuration at a critical attractive interaction strength. In correspondence of this phase transition the superfluid fraction strongly reduces and goes to zero for a sufficiently attractive Bose-Bose interaction.

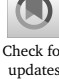
# 1 Introduction

How does a locally-varying spatial curvature influence the properties of low-dimensional quantum systems? This is a relevant question asked by scientists working in very different fields such as the linear Schrödinger equation for a particle constrained on a curve manifold [1–3], but also quantum gravity [4] or quantum chaos [5]. It is well know [6–8] that the local curvature of a curve on the three-dimensional (3D) Euclidean space is characterized by the so-called geodesic curvature. This geodesic curvature $\kappa$ is an extrinsic quantity: it does not remain invariant if the curve is under the effect of a distance-preserving transformation [6–8]. Instead, the local curvature of a surface on the 3D Euclidean space is characterized by the so-called Riemann curvature tensor, which can be written in terms of the invariant Gaussian curvature and the not-invariant average curvature [6–8]. The quantum motion of a particle on a curved waveguide has been anayzed by several authors [9–14]. More recently, the highly non-trivial role of curvature for constrained quantum systems has been theoretically investigated with ultracold atomic gases confined in a quasi-1D [15–18] and quasi-2D configurations [19]. The main result of all these investigations is that the local curvature gives rise to a quantum-curvature potential.

In this paper we consider an atomic Bose-Einstein condensate (BEC) confined in a quasi-1D elliptical waveguide finding that the quantum-curvature potential has the shape of a double-well, if the eccentricity of the ellipse is different from zero. By numerically solving the 1D Gross-Pitaevskii equation of the BEC wavefunction under the effect of this quantum-curvature potential, we show that the ground state of the system is uniform along the waveguide only if the eccentricity $\epsilon$ of the ellipse is zero (circular waveguide with constant curvature). Instead, for $\epsilon \neq 0$ we find that the ground state is generically characterized by a two-peak configuration, where the peaks are located around the minima of the effective double-well potential. However, we discover that in the case of attractive interaction it exists a critical (negative) interaction strength below which the ground state exhibits a quantum phase transition from the two-peak configuration to a one-peak configuration. This is the analog of the spontaneous symmetry breaking, i.e. the modulational instability [20], of the uniform configuration predicted some years ago for an 1D attractive BEC in a circular waveguide [21–23]. Our results show that the critical interaction strength depends on the eccentricity $\epsilon$ of the ellipse in a non-trivial way. We also analyze the effect of a boost velocity on the BEC moving in the elliptical waveguide deriving the Leggett formula [24] for the superfluid fraction of a 1D bosonic system [25–27]. Our numerical investigation reveals that the superfluid fraction decreases dramatically in response to this quantum phase transition, eventually reaching zero for a sufficiently negative Bose-Bose interaction.

# 2 Quantum-curvature potential

We consider a Bose-Einstein condensate (BEC) made of $N$ identical bosonic atoms of mass $m$. The atoms are constrained to move along a curve $\mathcal{C}$ by the presence of a strong harmonic potential of frequency $\omega_\perp$ in the local transverse plane with respect to $\mathcal{C}$. The characteristic length of the transverse confinement is $l_\perp = \sqrt{\hbar/(m\omega_\perp)}$ where $\hbar$ is the reduced Planck constant. We introduce a local system $(s, u, v)$ of coordinates, where $s$ is the curvilinar abscissa (arclength) along $\mathcal{C}$ while $u$ and $v$ are two coordinates of the transverse plane [15–18]. In this way the Lagrangian density of our problem is given by

$$\mathscr{L} = \frac{i\hbar}{2}(\Psi^* \partial_t \Psi - \Psi \partial_t \Psi^*) - \frac{\hbar^2}{2m}|\nabla\Psi|^2 - \frac{m\omega_\perp^2}{2}(u^2 + v^2)|\Psi|^2 - \frac{1}{2}g|\Psi|^4, \tag{1}$$

where $\Psi(s,u,v,t)$ is the BEC wavefunction normalized to one and $g = 4\pi\hbar^2 a_s(N-1)/m$ is the 3D strength of the contact inter-atomic potential with $a_s$ the s-wave scattering length. Clearly, the Laplacian operator $\nabla^2$ must be written in terms of the local system $(s,u,v)$ of coordinates [15–18]. Assuming the factorization

$$\Psi(s,u,v,t) = \psi(s,t)\,\frac{e^{-\frac{(u^2+v^2)}{2\sigma(s,t)^2}}}{\pi^{1/2}\sigma(s,t)} \tag{2}$$

and inserting this ansatz into the Lagrangian density, after integration over $u$ and $v$ one gets [18, 28, 29],

$$
\begin{aligned}
\mathscr{\bar{L}} =\ & \frac{i\hbar}{2}\left(\psi^*\partial_t\psi - \psi\partial_t\psi^*\right) - \frac{\hbar^2}{2m}|\partial_s\psi|^2 + \frac{\hbar^2\kappa^2(s)}{8m}|\psi|^2 \\
& -\left(\frac{\hbar^2}{2m}\frac{1}{\sigma^2} + \frac{m\omega_\perp^2}{2}\sigma^2\right)|\psi|^2 - \frac{1}{2}\frac{g}{2\pi\sigma^2}|\psi|^4,
\end{aligned}
\tag{3}
$$

where $\kappa(s)$ is the local geodesic curvature of $\mathcal{C}$, and the conditions $\sigma\kappa \ll 1$ and $\sigma \ll \xi$ must hold, with $\xi = \hbar/\sqrt{2g|\Psi|^2}$ the 3D healing length [18]. The Euler-Lagrange equations of the 1D action functional with respect to the 1D wavefunction $\psi(s,t)$ and the transverse width $\sigma(s,t)$ are

$$i\hbar\partial_t\psi = \left[-\frac{\hbar^2}{2m}\partial_s^2 - \frac{\hbar^2\kappa^2(s)}{8m} + \frac{\hbar^2}{2m}\frac{1}{\sigma^2} + \frac{m\omega_\perp^2}{2}\sigma^2 + \frac{2\hbar^2 a_s(N-1)}{m\sigma^2}|\psi|^2\right]\psi, \tag{4}$$

and

$$\sigma^2 = l_\perp^2\sqrt{1 + 2a_s(N-1)|\psi|^2}. \tag{5}$$

Eq. (4), equipped with Eq. (5), is the time-dependent 1D nonpolynomial Schrödinger equation (NPSE) [28, 29] for the wavefunction $\psi(s,t)$ of the BEC moving along the curve $\mathcal{C}$ (see also [18]). As previously discussed, the geodesic curvature $\kappa(s)$ gives rise to an effective potential

$$U_Q(s) = -\frac{\hbar^2\kappa(s)^2}{8m}. \tag{6}$$

This curvature potential $U_Q(s)$ is quantum because it involves the square of the reduced Planck constant $\hbar$. At fixed atomic mass $m$, only if the square of the curvature $\kappa(s)$ is sufficently large the effects of this quantum-curvature potential become relevant.

Under the assumption that $\sigma \simeq l_\perp$, which corresponds to a very strong transverse confinement, the 1D NPSE becomes the familiar 1D Gross-Pitaevskii (GPE) equation

$$i\hbar\partial_t\psi = \left[-\frac{\hbar^2}{2m}\partial_s^2 - \frac{\hbar^2\kappa(s)^2}{8m} + \hbar\omega_\perp + \frac{2\hbar^2 a_s(N-1)}{ml_\perp^2}|\psi|^2\right]\psi. \tag{7}$$

It is very important to stress that, from Eq. (5), the condition $\sigma \simeq l_\perp$ implies $2a_s(N-1)|\psi|^2 \ll 1$. In the rest of the paper we will work within this 1D regime. n the new version of the manuscript I shall discuss the role Clearly, Eq. (7) is reliable in the weak-coupling and strong-transverse-confinement regime, where both beyond-mean-field and transverse-size effects are very small.

## 3 Properties of the elliptical waveguide

We now choose an ellipse for the curve $\mathcal{C}$. By using cartesian coordinates its defining equation reads

$$\frac{x^2}{a^2} + \frac{y^2}{b^2} = 1, \tag{8}$$

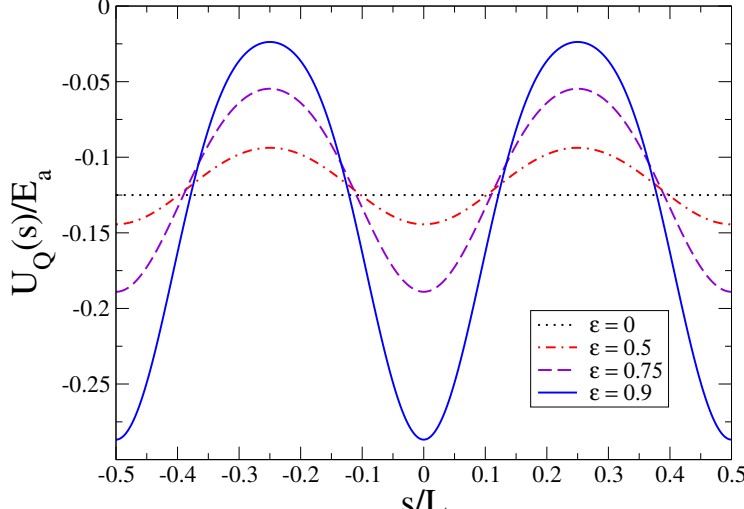

Figure 1: Quantum-curvature potential $U_Q(s)$, Eq. (6), induced by the geodesic curvature $\kappa(s)$ of an ellipse, as a function of the arclength $s$, where $a$ is the length of the major semi-axis, $L = aE(2\pi, \epsilon)$ is the perimeter of the ellipse, and $E_a = \hbar^2/(ma^2)$ a characteristic energy. The curves are obtained for different values of the eccentricity $\epsilon$.

where $a$ and $b$ are the lengths of the two semi-axes of the ellipse. Here we assume that $a \geq b$, such that $a$ is the length of the major semi-axis. The eccentricity of the ellipse is defined as [8]

$$\epsilon = \sqrt{1 - \frac{b^2}{a^2}}. \tag{9}$$

Clearly, $0 \leq \epsilon < 1$ and for $\epsilon = 0$ we obtain a circle of radius $R = a = b$. Introducing the angle $\phi \in [0, 2\pi]$ we can write

$$x = a \cos(\phi), \tag{10}$$
$$y = b \sin(\phi), \tag{11}$$

and the arclength $s$ along the ellipse can be expressed with the formula [8]

$$s = a E(\phi, \epsilon), \tag{12}$$

where

$$E(\phi, \epsilon) = \int_0^\phi \sqrt{1 - \epsilon^2 \sin^2(\phi')} \, d\phi' \tag{13}$$

is the incomplete elliptic integral of the second kind. It follows that the perimeter $L$ of the ellipse reads

$$L = a E(2\pi, \epsilon), \tag{14}$$

such that for $\epsilon = 0$ we have $L = 2\pi a$ because $E(2\pi, 0) = 2\pi$. Instead, for $\epsilon \to 1$ we have $L \to 4a$ because $E(2\pi, 1) = 4$. We conclude that $4a < L \leq 2\pi a$. The geodesic curvature $\kappa$ of the ellipse can be written as [8]

$$\kappa = \frac{1}{a} \frac{\sqrt{1 - \epsilon^2}}{\left(\sin^2(\phi) + \sqrt{1 - \epsilon^2} \cos^2(\phi)\right)^{3/2}}. \tag{15}$$

At fixed $a$ and $\epsilon$, the maximum of the curvature is obtained for $\phi = 0$ and $\phi = \pi$, i.e.

$$\kappa_{max} = \frac{1}{a} \frac{1}{(1-\epsilon^2)^{1/4}}, \tag{16}$$

while the minimum of the curvature is obtained for $\phi = \pi/2$ and $\phi = 3\pi/2$, i.e.

$$\kappa_{min} = \frac{1}{a} \sqrt{1-\epsilon^2}. \tag{17}$$

Thus, for $\epsilon = 0$ we have $\kappa_{max} = \kappa_{min} = 1/a$ while for $\epsilon \to 1$ we have $\kappa_{max} \to +\infty$ and $\kappa_{min} \to 0$. We conclude that $1/a \leq k_{max} < +\infty$ and $0 < k_{min} \leq 1/a$. In general, the formula which gives the curvature $\kappa$ as a function of the arclength $s$ is called Cesaro equation. Unfortunately, in the case of the ellipse there is no Cesaro equation. In other words, an analytical formula of $\kappa$ as a function of $s$ is not available. However, from Eqs. (12) and (15), fixing the length $a$ and the eccentricity $\epsilon$ of the ellipse, we can easily plot $\kappa$ vs $s$ using $\phi$ as dummy variable. More explicitly: we calculate separately $\kappa$ vs $\phi$ and $s$ vs $\phi$, and then we plot $\kappa$ vs $s$. The curvature $\kappa(s)$ has a the periodic structure of $\kappa(s)$. By increasing $\epsilon$, the perimeter $L$ of the ellipse slightly decreases while $\kappa_{max}$ and $\kappa_{min}$ pull away. In Fig. 1 we plot the quantum-curvature potential $U_Q(s)$, Eq. (6), induced by the curvature $\kappa(s)$ of the ellipse for different values of the eccentricity $\epsilon$. The figure clearly shows that, for $\epsilon \neq 0$, $U_Q(s)$ is symmetric double-well potential where the depth of the wells becomes larger by increasing the eccentricity. The minima (maxima) of the quantum-curvature potential $U_Q(s)$ are in correspondence to the maxima (minima) of the curvature $\kappa(s)$.

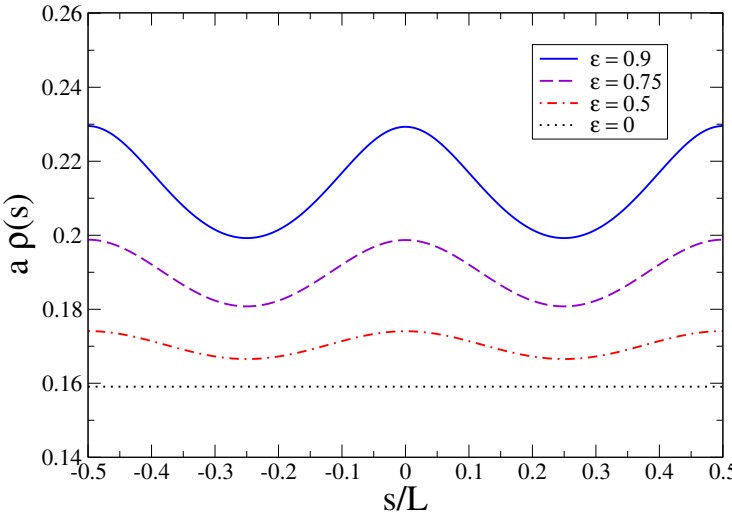

Figure 2: Probablility density $\rho(s)$ of the non-interacting BEC ground state in an ellipse as a function of the arclength $s$, where $a$ is the length of the major semi-axis and $L = aE(2\pi, \epsilon)$ is the perimeter of the ellipse. The curves are obtained for different values of the eccentricity $\epsilon$. Here the s-wave scattering length $a_s$ is set to zero or, equivalently, the number $N$ of particles is set to one.

## 4 BEC ground state in elliptical waveguide

The time-independent 1D GPE is obtained from Eqs. (7) setting

$$\psi(s,t) = \Phi(s) e^{-i(\mu + \hbar\omega_\perp)t/\hbar}. \tag{18}$$

In this way we have

$$\mu \Phi = \Big[ -\frac{\hbar^2}{2m}\partial_s^2 - \frac{\hbar^2 \kappa(s)^2}{8m} + \frac{2a_s(N-1)\hbar^2}{ml_\perp^2}|\Phi|^2 \Big]\Phi, \tag{19}$$

that is the 1D GPE equation for the stationary wavefunction $\Phi(s)$, such that $\rho(s) = |\Phi(s)|^2$ is the probability density of finding the BEC at the position $s$. In Fig. 2 we report the probability density $\rho(s)$ of the ground-state of the non-interacting ($a_s = 0$ or, equivalently, $N = 1$) BEC confined along the ellipse as a function of the arclength $s$. Our results are obtained by solving Eq. (7) with a Crank-Nicolson predictor-corrector method and imaginary time. In the figure, the curves correspond to different values of the eccentricity $\epsilon$. Clearly, for $\epsilon = 0$ the ground state is uniform along the ellipse. However, for $\epsilon \neq 0$ the ground state is no more uniform due to a non-constant curvature $\kappa(s)$ which implies a non-constant effective potential $U_Q(s) = -\hbar^2 \kappa(s)^2/(8m)$. By increasing the eccentricity $\epsilon$ the localization of $\rho(s)$ around the minima of $U_Q(s)$, where the curvature is larger, becomes more evident.

## 5 Quantum phase transition

It is interesting to investigate the effect of the inter-atomic interaction on the ground state properties of the system. In adimensional units the interaction strength reads $\gamma = g/(2\pi l_\perp^2 a E_a) = 2aa_s(N-1)/l_\perp^2$ with $E_a = \hbar^2/(ma^2)$. In Fig. 3 we consider the case of an attractive BEC and plot our numerical results obtained for different values of a negative $\gamma$ with fixed eccentricity $\epsilon = 0.9$. Quite remarkably, for $\gamma < -1.5$ the ground state has a spontaneous symmetry breaking: one of the two local minima contains more bosons. Indeed for $\gamma = -2$ this single-well localization is very clear. It is important to observe that this kind of quantum phase transition happens for any $\epsilon$. In the case of a circle ($\epsilon = 0$) a similar spontaneous symmetry breaking was predicted about 20 years ago [21–23]. Actually, this quantum

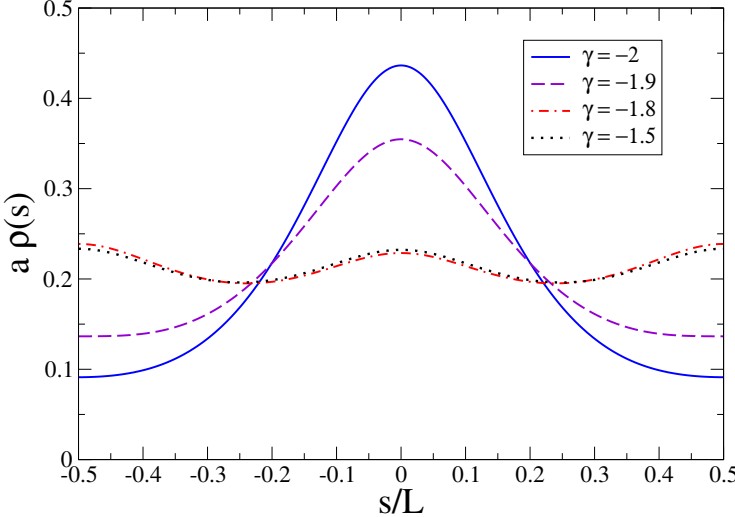

Figure 3: Probablility density $\rho(s)$ of the attractive BEC ground state in an ellipse as a function of the arclength $s$, where $a$ is the length of the major semi-axis and $L = aE(2\pi, \epsilon)$ is the perimeter of the ellipse. The curves are obtained with eccentricity $\epsilon = 0.9$ for different values of the adimensional interaction strength $\gamma = 2aa_s(N-1)/l_\perp^2$, where $a_s$ is the 3D s-wave scattering length and $l_\perp$ is the characteristic length of the transverse harmonic confinement.

phase transition, or spontaneous symmetry breaking, is nothing else than the modulational instability of the ground-state configuration, induced by the appearance of an imaginary component in the energies of the elementary excitations of the ground state [20]. However, for $\epsilon = 0$ there is a quantum phase transition from a uniform configuration to a single-peak configuration, while for the $\epsilon \neq 0$ there is a quantum phase transition from a two-peak configuration to a single-peak configuration. This quantum phase transition was observed some years ago with an attractive BEC of $^{39}$K atoms, where the double-well potential was created by intersecting two pairs of laser beams [31]. Our double-well system is slightly different because the particles tunnel from one well to the other well following two different curved paths; moreover, our elipsoidal configuration offers also the possibility of having persistent currents. For a sufficiently attractive BEC the transverse width $\sigma$ of the BEC becomes smaller than $l_\perp$ and there is the collapse of the single-peak configuration. For $\epsilon = 0$ the 1D NPSE predicts the collapse of this single-peak configuration at $\gamma_c = (4/3)(a/l_\perp)$ [28, 30]. Thus, our numerical results of Fig. 3, obtained from the 1D GPE, are fully reliable under the condition $a/l_\perp \gg 1$. This is again the condition of a tight transverse confinement.

## 6 Superfluid fraction

Let us consider the effect of a boost velocity $v_B$ on the BEC moving along the ellipse. In this case Eq. (19) is modified as follows

$$\mu\, \Phi = \left[ \frac{1}{2m}(-i\hbar\partial_s - mv_\mathrm{B})^2 - \frac{\hbar^2\kappa^2(s)}{8m} + \frac{2a_s(N-1)\hbar^2}{ml_\perp^2}|\Phi|^2 \right]\Phi\,, \qquad (20)$$

Now we set

$$\Phi(s) = \frac{n(s)^{1/2}}{\sqrt{N}}\, e^{i\theta(s)}\,, \qquad (21)$$

where $n(s) = N\rho(s)$ is the local number density of the BEC. Morover, we introduce the local velocity field

$$v(s) = \frac{\hbar}{m}\partial_s\theta(s)\,. \qquad (22)$$

Inserting these formulas into Eq. (20) we obtain 1D stationary equations of zero-temperature superfluid hydrodynamics

$$\mu = -\frac{\hbar^2}{2m\sqrt{n}}\partial_s^2\sqrt{n} + \frac{m}{2}(v - v_\mathrm{B})^2 - \frac{\hbar^2\kappa(s)^2}{8m} + \frac{2a_s(1-1/N)\hbar^2 n}{ml_\perp^2}\,, \qquad (23)$$

and also

$$\partial_s\left[ n(v - v_B) \right] = 0\,. \qquad (24)$$

Eq. (24) implies that $n(v - v_B) = J$, where $J$ is a constant current density. This result is very interesting because it says that if $n(s)$ has spatial variations then also $v(x)$ must have spatial variations.

Inspired by Ref. [27], we now introduce the average value of the velocity $v(x)$ in a region $[a, b]$ of the ellipse as

$$\bar{v} = \frac{1}{(b-a)}\int_a^b v(s)\, ds\,, \qquad (25)$$

Then, from the previous equations we obtain

$$\bar{v} = \frac{1}{(b-a)}\int_a^b \left( \frac{J}{n(s)} + v_B \right) ds = \frac{J}{\bar{n}_s} + v_B\,, \qquad (26)$$

where

$$\bar{n}_s = \frac{1}{\frac{1}{(b-a)} \int_a^b \frac{1}{n(s)} ds} \, .$$

(27)

The number density $\bar{n}_s$ can be interpreted as the superfluid number density of the stationary state in the spatial region $[a, b]$. Indeed, Eq. (27) is the 1D version of the formula obtained by Leggett [24] for a supersolid with spatial periodicity $(b - a)$, and recently discussed by others [25–27]. If the stationary state $\Psi(s)$ moves with the average velocity $\bar{v}$, its current density reads $J = \bar{n}_s (\bar{v} - v_B)$, where $\bar{v}$ is the average velocity in the region $[a, b]$ and $\bar{n}_s$ the corresponding superfluid number density. We can also introduce

$$\bar{n} = \frac{1}{(b-a)} \int_a^b n(s) \, ds$$

(28)

that is the average number density in the region $[a, b]$. Consequently, the superfluid fraction of the BEC in the region $[0, L]$ reads

$$f_s = \frac{\bar{n}_s}{\bar{n}} = \frac{1}{\frac{N}{L^2} \int_0^L \frac{1}{n(s)} ds} \, ,$$

(29)

where $N = \int_0^L n(s) \, dx = L\bar{n}$. This formula can be also obtained as the response of the linear momentum of the BEC to the boost velocity $\bar{v}_B$, that is the non-classical translational inertia of the system [25].

In Fig. 4 we plot our numerical results of the superfluid fraction $f_s$ as a function of the adimensional interaction strength $\gamma$ for different values of the eccentricity $\epsilon$ of the ellipse. For positive values of $\gamma$ the superfluid fraction $f_s$ is close to 1 also with $\epsilon = 0.9$. However, for $0 \le \gamma < 1$ and a very large eccentricity ($\epsilon = 0.99$) we find $f_s \simeq 0.95$. This result is quite reasonable because the wavefunction is strongly localized in the two well of the quantum-curvature potential. For negative values of $\gamma$ the most interesting effect appears in Fig. 4:

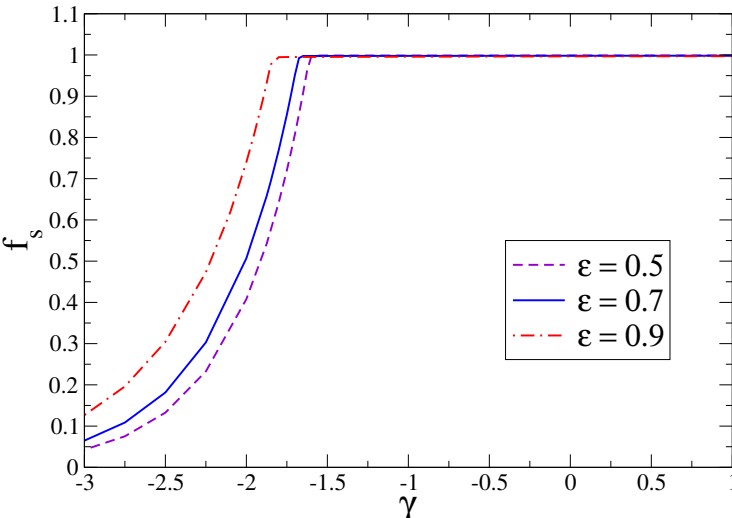

Figure 4: Superfluid fraction $f_s$ of the BEC ground state in an ellipse as a function of the adimensional interaction strength $\gamma = 2aa_s(N-1)/l_\perp^2$, where $a$ is the length of the major semi-axis, $a_s$ is the 3D s-wave scattering length and $l_\perp$ is the characteristic length of the transverse harmonic confinement. The curves are obtained for three values of the eccentricity $\epsilon$ of the ellipse.

around $\gamma \simeq -1.6$ the superfluid fraction $f_s$ quckly decreases and it goes to zero for very large negative values of $\gamma$. This is exacly the quantum phase transition from a two-peak configuration to a one-peak configuration. As previously stressed, the one-peak configuration becomes modulationally unstable when at least one of the energies of its elementary excitations acquires an imaginary component. A similar modulational instability [20] happens in the formation of a train of bright solitons from a single-peak Bose-Einstein condensate, induced by a sudden change in the sign of the scattering length from positive to negative [32]. Our Fig. 4 reveals that the critical strength $\gamma_c$ crucially depends on the eccentricity $\epsilon$, such as the behavior of $f_s$ as a function of $\gamma$ for $\gamma < \gamma_c$. In particular, we find that $\gamma_c$ slightly reduces by increasing $\epsilon$, but this effect is quite weak.

An important remark is that Eq. (29) has been derived here without any assumption about the sign of $\gamma$. Moreover, the absence of superfluidity for $\gamma = 0$ is true only in the thermodynamic limit. In a ring there is a finite energy gap between the ground state and the first excited state also for $\gamma = 0$. The Bose system we are considering has a finite size because it is confined in a finite elliptical ring. Finally, the 3D version of Eq. (27) was proposed historically by Leggett to characterize the superfluid density of a supersolid [24], while here Eq. (27) is used to determine the superfluid density of a Bose-Einstein condensate which is not supersolid but it is instead spatially modulated due to the crucial interplay between elliptical confinement and attractive interaction.

## 7 Conclusions

The main goal of this paper was to understand the role of a locally-varying curvature for a Bose-Einstein condensate confined in an elliptical waveguide. The proposed setup, and the double-well quantum-curvature potential that we have found, can be experimentally achieved by using ultracold atoms, which are a paradigmatic physical platform due to the high experimental tunability of inter-atomic interactions and trapping potentials. For instance, one can trap $N = 10^4$ ultracold Rb atoms by using a rapidly moving laser beam which creates a time-averaged elliptic-shaped toroidal optical dipole potential [33]. The length $a$ of the major semi-axis of the ellipse can be $a \simeq 100$ microns and the transverse length $l_\perp = 5$ microns. The scattering length $a_s$ could be then tuned by using an external magnetic field, which induces a Fano-Feshbach resonance [34]. Despite the fact that we focused on space curvature rather than space-time curvature, we believe that our results can be of interest not only to atomic and condensed matter physics researchers, but also to a large community working on general relativity and relativistic quantum field theory.

## Acknowledgements

The author thanks Francesco Ancilotto, Koichiro Furutani, Francesco Minardi, and Andrea Tononi for useful discussions.

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
