# Peer review of "Bose-Einstein condensate in an elliptical waveguide"

_SciPost Physics Core, doi:SciPost Phys. Core 5, 015 (2022)_

## Round 1 · Referee Report · Anonymous (Referee 3) · 2021-11-4

Report

This paper presents a theoretical study of a Bose-Einstein condensate confined in a quasi-uni-dimensional elliptic wave guide. The ellipticity of the guide induces an effective potential which is a periodic double well. The ground state is determined by numerically solving the mean field Gross-Pitaevskii equation. A quantum phase transition is identified in which most of the condensate spontaneously locates into one of the two identical minima of the potential.

Although the setting studied is elegant and may enable to study interesting aspects of the interplay between Bose condensation, superfluidity and an additional spontaneous symmetry breaking, I do not consider that this work presents enough new and interesting material.

(1) In particular, the fact that the author uses only a mean field approach is, according to me, a drawback, considering that beyond mean field corrections have already been addressed long ago in similar settings (Refs. [12] and [13] for instance).

(2) Also, I am skeptical concerning the use of Leggett's formula for determining the superfluid fraction in the absence of interatomic repulsion (i.e. when gamma=<0).
2a. It is known for instance that a uniform non interacting BEC is not superfluid, whereas Fig. 4 seems to indicate that the system is 100% supersolid when gamma=0 (i.e., in the absence of interaction).
2b. In the same line: at T=0 and gamma=0 the uniform system reaches a quantum tricritical point and the existence of a superfluid phase for negative s-wave scattering length depends on the contribution of higher order terms [see e.g., Zwerger, J. Stat. Mech., 103104 (2019)].
For these reasons, I think that the results presented in Fig. 4 should be considered with caution, or at least require a more extended discussion.

Considering the comments (1) and (2) above, I do not believe that the quality of this work warrants publication in SciPost Physics.

  • validity: -
  • significance: -
  • originality: -
  • clarity: -
  • formatting: -
  • grammar: -

Author:  Luca Salasnich  on 2021-11-21  [id 1964]

(in reply to Report 3 on 2021-11-04)
Category:
remark
objection

I thank the referee who thinks that the setting studied is elegant and may enable to study interesting aspects of the interplay between BEC and superfluidity. In the new version of the manuscript I shall discuss the role of beyond-mean-field corrections (also in the case \gamma=0): the calculations are obtained in a regime where both beyond-mean-field and transverse- size effects are very small. Regarding the absence of superfluidity for \gamma = 0, this is true in the thermodynamic limit. However, in a ring there is a finite energy gap between the ground state and the first excited state also for \gamma = 0. In the new version of the paper I shall discuss this relevant issue related to the fact that the Bose system I am considering has a finite size because it is confined in a finite elliptical ring.

---

## Round 1 · Referee Report · Anonymous (Referee 2) · 2021-11-4

Report

The paper reports sufficiently interesting results which may draw interest of readers, and are recommended for the publication, provided that some amendments will be made, as summarized below.

A technical comment is that the underlying model is not formulated in a sufficiently clear form. Namely, in Eq. (7) the effective potential is written in terms of \kappa, which is defined in Eq. (15) as a function of \phi, but \phi is not defined as a function of coordinate s. Rather, s is defined as a function of \phi by Eqs. (12) and (13). Taken together, these implicit definitions seem confusing.

An essential comment is about what is called quantum phase transitions in the manuscript. First, in the case of the zero eccentricity, this transition is nothing else but the commonly known onset of the modulational instability in the NLS equation with periodic boundary conditions, in the absence of an external potential. It seems somewhat strange that MI is not mentioned, and the commonly known threshold for the onset of MI on the circular ring is not referred to. The main result, reported in the paper, viz., the transition to the single density peak in the elliptic ring, i.e., spontaneous symmetry breaking of the ground state in the dual-well potential, is quite interesting, but it will be relevant too to state how the ellipticity affects the onset of the MI on the ring.

  • validity: high
  • significance: good
  • originality: high
  • clarity: good
  • formatting: excellent
  • grammar: -

Author:  Luca Salasnich  on 2021-11-21  [id 1963]

(in reply to Report 2 on 2021-11-04)
Category:
remark
reply to objection

I thank the Referee. In the new version of the paper I shall try to better explain how to calculate the
effective potential: the plot \kappa vs s is obtained calculating \kappa vs \phi and also s vs \phi.
\phi is a dummy variable here. Unfortunately, an explicit analytical formula of \kappa as a function of s is not available. In the new version I shall also explain the relevant role of the modulational instability (of the Bogoliubov elementary exacitations) which triggers the spontaneous symmetry breaking.

---

## Round 1 · Referee Report · Anonymous (Referee 1) · 2021-11-4

Report

The paper reports sufficiently interesting results which may draw interest of readers, and are recommended for the publication, provided that some amendments will be made, as summarized below.

A technical comment is that the underlying model is not formulated in a sufficiently clear form. Namely, in Eq. (7) the effective potential is written in terms of \kappa, which is defined in Eq. (15) as a function of \phi, but \phi is not defined as a function of coordinate s. Rather, s is defined as a function of \phi by Eqs. (12) and (13). Taken together, these implicit definitions seem confusing.

An essential comment is about what is called quantum phase transitions in the manuscript. First, in the case of the zero eccentricity, this transition is nothing else but the commonly known onset of the modulational instability in the NLS equation with periodic boundary conditions, in the absence of an external potential. It seems somewhat strange that MI is not mentioned, and the commonly known threshold for the onset of MI on the circular ring is not referred to. The main result, reported in the paper, viz., the transition to the single density peak in the elliptic ring, i.e., spontaneous symmetry breaking of the ground state in the dual-well potential, is quite interesting, but it will be relevant too to state how the ellipticity affects the onset of the MI on the ring.

---

## Round 2 · Referee Report · Anonymous (Referee 1) · 2022-1-26

Report

The revision of the paper is adequate. Th resubmitted paper may be recommended for the publication.
  • validity: top
  • significance: high
  • originality: high
  • clarity: high
  • formatting: excellent
  • grammar: excellent

Author:  Luca Salasnich  on 2022-01-28  [id 2127]

(in reply to Report 1 on 2022-01-26)

I thank the Referee who thinks that the revision is adequate.

---

## Round 2 · Referee Report · Anonymous (Referee 3) · 2022-1-28

Report

This is a beautiful work considering the effect of curvature in the NPSE derived by the author and collaborators years ago. Specifically, when a quantum matter-wave is embedded in a vent waveguide under tight-transverse confinement, it is possible to integrate out the transverse degrees of freedom and reduce the dimensionality of the description. Curvature effects can be captured by potential terms along the arc length. In the simplest case, one finds a potential proportional to the curvature square. The generalization of these results to the NLSE was studied earlier on by others, such [6,7]. In parallel, the NPSE was introduced as an improvement for the dimensional reduction of the NLSE in tight waveguides in the absence of curvature. From that point of view, the current study is a natural step forward.

The work is nicely written but the bibliography does not make justice to the abundant literature on curvature-induced effects. Much of this is indeed focused on the linear case, but this is still a reference limit in the present study. I encourage the author to put the work into a broader context, making a minor revision, citing the pioneering and beautiful early works

Switkes, E., Russel, E. L. & Skinner, J. L. Kinetic energy and path curvature in bound state systems. J. Chem. Phys. 67, 3061 (1977).
da Costa, R. C. T. Quantum mechanics of a constrained particle. Phys. Rev. A 23, 1982 (1981).
da Costa, R. C. T. Constraints in quantum mechanics. Phys. Rev. A 25, 2893 (1982).
Goldstone, J. & Jaffe, R. L. Bound states in twisting tubes. Phys. Rev. B 45, 14100 (1992).

In the math-phys literature, other authors like Bracken and Exner devoted a great deal of attention to the linear case

Clark, I. J. & Bracken, A. J. Effective potentials of quantum strip waveguides and their dependence upon torsion. J. Phys. A: Math. Gen. 29, 339 (1996).
Clark, I. J. More on effective potentials of quantum strip waveguides. J. Phys. A: Math. Gen. 31, 2103 (1998).
Exner, P. & Seba, P. Bound states in curved quantum waveguides. J. Math. Phys. 30, 2574 (1989).
Exner, P. & Vugalter, S. A. On the number of particles that a curved quantum waveguide can bind. J. Math. Phys. 40, 4630 (1999).

I believe there is even a book by Exner on the topic. Here it is:
Quantum Waveguides. Pavel Exner, Hynek Kovařík (2015)

With an eye on applications, it is my impression that curvature-induced effects still play a subdominant role in BEC/atomtronics given the scales involved. One can get a feeling considering the length scales required for the da CIP/Costa potential to affect the dynamics. Still, that is a purely technological issue that should not prevent this kind of study.

If the author is having fun with this, an interesting prospect is the inclusion of torsion effects that have been much less studied, to the best of my knowledge.

Beyond that, I think the manuscript makes a valuable contribution that is nicely elaborated and worth publishing.
  • validity: high
  • significance: high
  • originality: high
  • clarity: high
  • formatting: good
  • grammar: good

Author:  Luca Salasnich  on 2022-01-28  [id 2126]

(in reply to Report 2 on 2022-01-28)

Thanks a lot for the nice Report. I will be delighted to improve the paper by adding and discussing the relevant references suggested by the Referee.

---

## Round 2 · Author Response

Response to Referee 1 (and 2)

I thank the Referee for the useful comments and suggestions.

In the new version of the paper I tried to better explain how to calculate the
effective potential: the plot \kappa vs s is obtained calculating \kappa vs \phi and also s vs \phi.
\phi is a dummy variable here. Unfortunately, an explicit analytical formula of \kappa as a function of s is not available.

In the new version I also explained the relevant role of the modulational instability (of the Bogoliubov elementary excitations) which triggers the spontaneous symmetry breaking.

The changes related to the suggestions and requests of Referee 1 (and 2) are in blue.

Response to Referee 3

I thank the referee, who thinks that the setting studied is elegant and may enable to study interesting aspects of the interplay between BEC and superfluidity.

In the new version of the manuscript I better discussed the absence of beyond-mean-field corrections in the model: the calculations are obtained in a regime where both beyond-mean-field and transverse- size effects are very small.

Regarding the absence of superfluidity for \gamma = 0, this is true in the thermodynamic limit. However, in a ring there is a finite energy gap between the ground state and the first excited state also for \gamma = 0. In the new version of the paper I emphasized this relevant issue: the Bose system I am considering has a finite size because it is confined in a finite elliptical ring.

The changes related to the suggestions and requests of Referee 3 are in red.

---

## Round 2 · List of Changes

1. In the Introduction I added a comment about the modulational instability citing a new reference, Ref. [11].

  2. At the end of Section III there is now a phrase about the fact that Eq. (7) is reliable only in the weak-coupling and strong-transverse-confinement regime.

  3. At the end of Section IV there is a more detailed explanation about the determination of the plot of \kappa vs s. The key idea is to use \phi as dummy variable.

  4. At the beginning of Section VI there is a discussion of the equivalence, in our specific problem, of quantum phase transition, spontaneous symmetry breaking, and modulational instability.

  5. After Eq. (29) I included another discussion about the modulational instability, quoting a new reference, Ref. [23]. Moreover, it has been added a sentence about the dependence of \gamma_c with respect to \epsilon.

  6. At the end of Section VI there are some arguments about the validity of Eqs. (27) and (29) for our finite-size system.

---

## Round 3 · Author Response

Response to Report 1.
I thank the Referee who thinks that the revision is adequate.

Response to Report 2.
Thanks a lot for the nice Report. I have improved the paper by adding and discussing the relevant references suggested by the Referee. The changes are in blue.

---

## Round 3 · List of Changes

1. I have changed and added some sentences in the Introduction.
  2. I have added 8 references.

All the changes are in blue.

---

## Editorial Decision

published